# MotorNet, a Python toolbox for controlling differentiable biomechanical effectors with artificial neural networks

**Olivier Codol[1,2]\*, Jonathan A Michaels[1,3,4], Mehrdad Kashefi[1,3,4], J Andrew Pruszynski[1,2,3,4], Paul L Gribble[1,2,3]**

[1]Western Institute for Neuroscience, University of Western Ontario, Ontario, Canada; [2]Department of Psychology, University of Western Ontario, Ontario, Canada; [3]Department of Physiology & Pharmacology, Schulich School of Medicine & Dentistry, University of Western Ontario, Ontario, Canada; [4]Robarts Research Institute, University of Western Ontario, Ontario, Canada

**Abstract** Artificial neural networks (ANNs) are a powerful class of computational models for unravelling neural mechanisms of brain function. However, for neural control of movement, they currently must be integrated with software simulating biomechanical effectors, leading to limiting impracticalities: (1) researchers must rely on two different platforms and (2) biomechanical effectors are not generally differentiable, constraining researchers to reinforcement learning algorithms despite the existence and potential biological relevance of faster training methods. To address these limitations, we developed MotorNet, an open-source Python toolbox for creating arbitrarily complex, differentiable, and biomechanically realistic effectors that can be trained on user-defined motor tasks using ANNs. MotorNet is designed to meet several goals: ease of installation, ease of use, a high-level user-friendly application programming interface, and a modular architecture to allow for flexibility in model building. MotorNet requires no dependencies outside Python, making it easy to get started with. For instance, it allows training ANNs on typically used motor control models such as a two joint, six muscle, planar arm within minutes on a typical desktop computer. MotorNet is built on PyTorch and therefore can implement any network architecture that is possible using the PyTorch framework. Consequently, it will immediately benefit from advances in artificial intelligence through PyTorch updates. Finally, it is open source, enabling users to create and share their own improvements, such as new effector and network architectures or custom task designs. MotorNet's focus on higher-order model and task design will alleviate overhead cost to initiate computational projects for new researchers by providing a standalone, ready-to-go framework, and speed up efforts of established computational teams by enabling a focus on concepts and ideas over implementation.

**\*For correspondence:**
codol.olivier@gmail.com

## eLife assessment

This work will be of interest to the motor control community as well as neuroAI researchers interested in how bodies constrain neural circuit function. The authors present "MotorNet", a **useful** software package to train artificial neural networks to control a biomechanical model of an effector. The manuscript provides **solid** evidence that MotorNet is easy to use and can reproduce past results in the field, both at the neural and behavioural levels. Validation is limited to planar arm-like plants or point-masses, so future work exploring three-dimensional movements and other types of plants would strengthen the impact of the tool.

## Introduction

Research on the neural control of movement has a long and fruitful history of complementing empirical studies with theoretical work (*Lindsay, 2022*). Consequently, a wide variety of computational model classes have been proposed to explain empirical observations, such as equilibrium point control (*Feldman and Levin, 1995*; *Flanagan et al., 1993*; *Gribble and Ostry, 2000*; *Won and Hogan, 1995*), optimal control (*Shadmehr and Krakauer, 2008*; *Todorov, 2004*), and parallel distributed processing models (*Fetz, 1993*; *Gomi and Kawato, 1993*; *Jordan and Rumelhart, 1992*; *Lillicrap and Scott, 2013*), commonly known as artificial neural networks (ANNs). Although ANNs were formalized many decades ago, they gained in popularity only recently following their rise to prominence in machine learning (ML; *LeCun et al., 2015*), as their greater explanatory power and biological realism provide significant advantages against alternative model classes (*Gershman and Ölveczky, 2020*; *Lillicrap et al., 2019*; *Richards et al., 2019*; *Saxe et al., 2021*).

For neural control of movement, production of theoretical work using ANN models may be viewed as a two-step effort: (1) building a realistic simulation environment that mimics the behaviour of bodily effectors and (2) implement the policy ANNs themselves to train on the environment. Many open-source platforms achieve each of these steps individually, such as MuJoCo (*Todorov et al., 2012*) or OpenSim (*Delp et al., 2007*; *Seth et al., 2018*) for building environments, and JAX, PyTorch, or TensorFlow for building and training policy ANNs. However, approaches using these platforms lead to two important impracticalities.

First, the user must rely on two different software platforms, one for the environment and one for the policy ANN. Communication between platforms is not built-in, requiring users to produce custom code to link the policy ANN software with the software implementing the simulation of the environment. This forces significant overhead cost to initiate computational projects and creates barriers to research teams who lack the technical background to build those custom pipelines. A current remedy to this issue is *gymnasium* (*Chinnaiya et al., 2023*), a Python toolbox that provides an interface between policies and environments.

However, *gymnasium* constrains the user to reinforcement learning algorithms (*Fujimoto et al., 2018*; *Lillicrap et al., 2019*; *Mnih et al., 2015*) despite the existence and potential biological relevance of faster training methods such as backpropagation (*Lillicrap et al., 2020*; *Whittington and Bogacz, 2017*). The inability to use backpropagation to train policies represents the second impracticality. To date, this has been circumvented by training separate ANNs such as multi-layer perceptrons or recurrent neural networks (RNNs) as 'forward models' approximating the behaviour of effectors that are normally implemented in a separate software package (e.g., *Lillicrap and Scott, 2013*; *Willett et al., 2021*). This approach does not address the need for custom pipelines, and remains a slow, cumbersome process when iterating over many different policies and environments, because new approximator ANNs must be trained each time.

Solving the issues described above requires both the policy and environment to rely on the same software (no-dependency requirement), and for the environment to allow for backpropagation through itself (differentiability requirement) so that typical gradient-based algorithms may be employed. Ideally, the solution would also be open source, modular for flexibility of coding and focus on ideas, and have reasonable training speeds on commercially available computers.

We developed MotorNet with these principles in mind. MotorNet is a freely available open-source Python toolbox (https://motornet.org) that allows for the training of ANNs to control arbitrarily complex and biomechanically realistic effectors to perform user-defined motor tasks. The toolbox requires no dependency besides standard Python toolboxes available on *pip* or Anaconda libraries. This greatly facilitates its use on remote computing servers as no third-party software needs to be installed. The environments are fully differentiable, enabling fast and efficient training of ANNs using standard gradient-based methods. It is designed with ease of installation and ease of use in mind, with a high-level and user-friendly application programming interface (API). Its programming architecture is modular to allow for flexibility in model building and task design. Finally, MotorNet is built on PyTorch, which makes innovation in ML readily available for use by MotorNet as they are implemented and released by the PyTorch Foundation. Here, we focus on illustrating the scientific use and relevance of the toolbox (the *why*), rather than the underlying API through coding snippets (the *how*), as the latter is more efficiently showcased via interactive, easily updatable online tutorials. The interested reader may consult the full API documentation, including interactive tutorials on the toolbox website at https://motornet.org.

# Results

## Training an ANN to perform a centre-out reaching task against a curl field

A canonical experimental paradigm in the study of neural control of movement is the centre-out reaching task with a 'curl field' applied at the arm's endpoint by a robot arm (*Conditt et al., 1997*; *Shadmehr and Mussa-Ivaldi, 1994*). In this paradigm, visual targets are placed around a central starting position in a horizontal plane. Participants must move the handle of a robot arm from the starting position to the target that appears on a given trial. During the reaching movement, the robot applies forces at the handle that scale linearly with the velocity of the hand and push in a lateral direction. This leads the central nervous system to adapt by modifying neural control signals to muscles to apply opposite forces to counteract and nullify the lateral forces produced by the robot. Finally, removal of the curl field leaves an opposite after-effect (*Shadmehr and Mussa-Ivaldi, 1994*). This paradigm is well suited to assess the functionality of MotorNet because it is well understood and

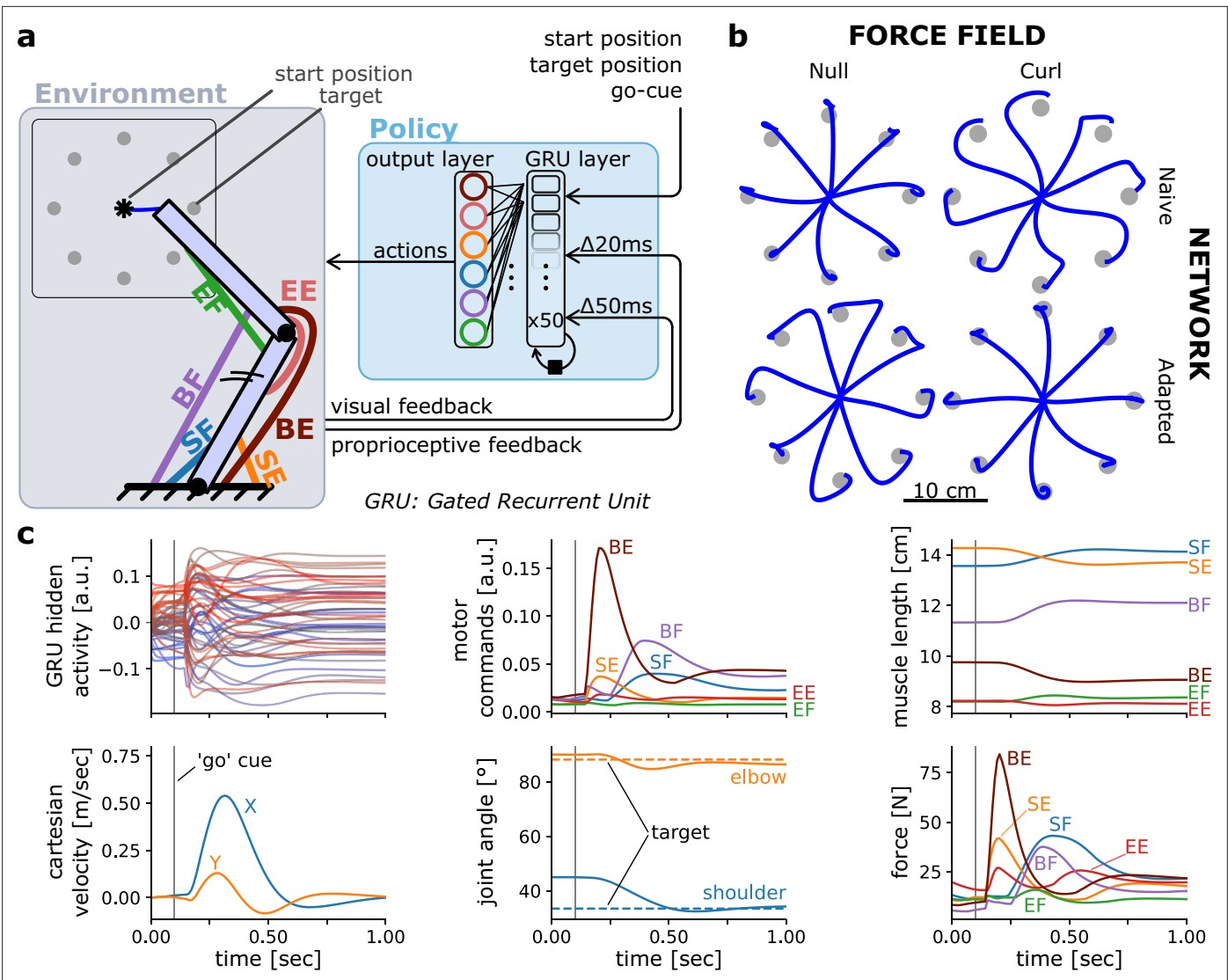

**Figure 1.** Controlling an arm-like effector in a centre-out reaching task with a curl field. (**a**) Schematic of the environment (containing the effector) and policy. (**b**) Endpoint trajectories of centre-out reaching movements in a null and curl field, for a policy recurrent neural network (RNN) that is untrained (naive) and then trained to reach in that curl field. The effector was as defined in *Kistemaker et al., 2010*. (**c**) Different variables over time during a rightward reaching movement.

extensively documented, and highlights physical, biomechanical, and control properties of human behaviour.

We specified a one-layer RNN composed of 50 gated recurrent units (GRUs; *Cho et al., 2014*) to control a two degrees of freedom (DoF), six muscle planar arm model (*arm26*; *Figure 1a*; *Kistemaker et al., 2006*; *Kistemaker et al., 2010*). The muscles were rigid-tendon, Hill-type muscle models, with 'shoulder' mono-articular flexors/extensors, 'elbow' mono-articular flexors/extensors, and a bi-articular pair of muscles producing flexion or extension at both joints (see Methods sections 'Arm26 model' and 'Model').

Training the model above took about 13 min on a 2022 Mac Studio with an M1 Max central processing unit (Apple Inc, Cupertino, CA, USA). Because the *arm26* effector and the centre-out reaching task are particularly common in the motor control literature, they are included in the toolbox as pre-built objects. Consequently, one can re-create the effector instance and the corresponding environment in one line of code for each. Note however that users can easily declare their own custom-made effector and environment objects if desired by subclassing the base *Effector* and *Environment* class, respectively (see below for more details on base classes and subclassing).

Including the implementation of the policy RNN and training routine, the above example can be reproduced with a few lines of code (see tutorial notebooks online), illustrating the ease of use of MotorNet's API. Once the model is trained, it can produce validation results via a forward pass (*Figure 1b, c*), which can then be saved and analysed afterwards. The results the model produces include joint and cartesian states (positions, velocities), muscle states (lengths, velocities, activations, forces), musculo-tendon states (lengths, velocities), efferent actions (i.e., time-varying muscle drive), and afferent feedback responses (proprioceptive, visual), as well as any activity states from the network if applicable (*Figure 1c*). Note that actions are different from muscle activations, in that they are input signals to the ordinary differential equation that produces muscle activation (see Methods; *Millard et al., 2013*; *Thelen, 2003*).

## Structure of MotorNet

Functionally, a MotorNet model can be viewed as a differentiable environment that can directly employ outputs from a policy ANN as action signals. The environment contains an effector, which is actuated according to the action input and in turn outputs information that may be fed back to the ANN (*Figure 2*). This closed-loop cycle repeats for each timestep. By default, 'visual' feedback consists of a vector indicating endpoint cartesian coordinates, while 'proprioceptive' feedback

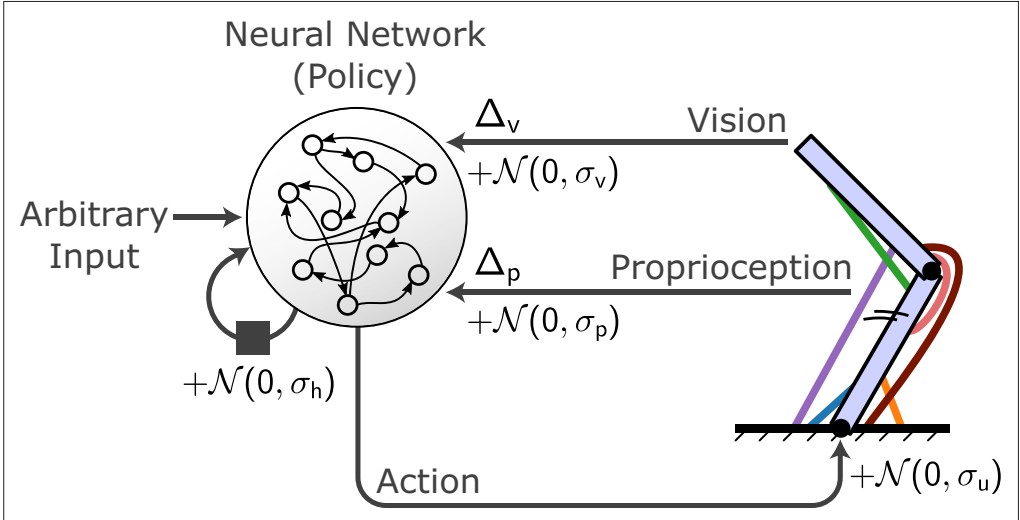

**Figure 2.** Conceptual organization of a MotorNet model. A policy artificial neural network (ANN) receives arbitrary input as well as recurrent connections from itself and sends action signals to an effector embedded in an environment, which in turn sends sensory feedback information. Typically, this feedback will be visual and proprioceptive, and can contain feedback-specific time delays $\Delta_p$ and $\Delta_v$. Gaussian noise can be added to the recurrent connection, action signal, and proprioceptive and visual feedback, with specific standard deviation $\sigma_h$, $\sigma_u$, $\sigma_p$, and $\sigma_v$.

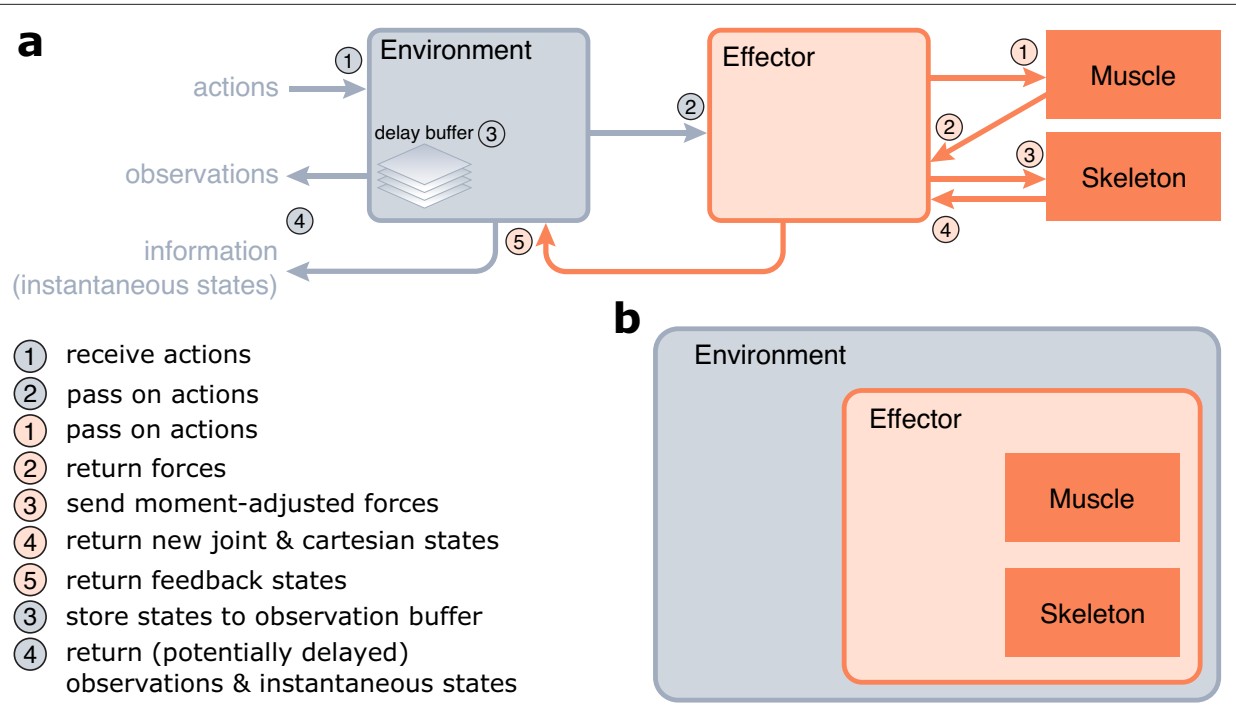

**Figure 3.** Implementation of MotorNet. (**a**) Information flow of a MotorNet model during runtime. (**b**) Declarative structure of a MotorNet object. Each object instance is held in memory as an attribute of another according to this hierarchical representation, except for the Muscle, and Skeleton instances.

consists of a $2m$-element vector of muscle length and velocity, with $m$ the number of muscles of the effector. Noise may be added in various parts of the model, such as on descending action signals or on feedback signals. Finally, time delays may be added to feedback signals before they reach the policy ANN. Importantly, the policy ANN may be any PyTorch network, and the MotorNet environments are designed to match standard *gymnasium* API conventions. That is, it is not necessary to create a policy by sub-classing a MotorNet *Policy* object.

## Running flow

At runtime, a more detailed representation of the information flow best describes how a MotorNet model behaves (*Figure 3a*). Models are based on five object classes: *Skeleton*, *Muscle*, *Effector*, *Environment*, and *Policy* objects (*Table 1*). Each object has its own base class, from which the user can create a custom subclass if desired. MotorNet comes with a set of pre-built subclasses for each, which implement commonly used computational model formalizations (*Table 1*).

*Environment* objects are the entry point of the model (*Figure 3a*). They take arbitrary action inputs, which are then passed on to the *Effector* object. *Effector* objects are essentially wrapper objects that hold the *Muscle* and *Skeleton* objects and handle coordination of information flow between them (*Figure 3a, b*), as well as concomitant numerical integration to ensure numerical stability. They pass the action signals to the *Muscle* object, which computes forces in return. The *Effector* will adjust those forces using geometry-dependent moment arms (see section 'Biomechanical properties of the effector' for details) and send the resulting generalized forces to the *Skeleton* object. These generalized forces will actualize the *Skeleton*'s joint state, which the *Skeleton* will return to the *Effector* object alongside the equivalent cartesian state. The *Effector* will then return the actuated states to the *Environment* object.

Finally, the *Environment* object will return an observation vector that contains arbitrarily processed information about the states of the *Environment* and *Effector* objects. These can usually be passed on to the policy ANN as input to perform the next forward pass. The *Environment* may maintain a delay buffer, which stores state information for a certain time (according to the $\Delta_p$ and $\Delta_v$ parameters, *Figure 2*), allowing the observation vector to be fed time-delayed state information. The *Environment* also outputs an information dictionary, which contains all the instantaneous (i.e., non-delayed) states

**Table 1.** Overview of Python base classes and their respective pre-built subclasses in MotorNet.
GRU: gated recurrent unit.

|  | Subclass | Description |
|---|---|---|
| Skeleton | *PointMass* | A skeleton with one bone of null length evolving in a plane. |
|  | *TwoJointArm* | A planar, two-segment skeleton with one hinge joint between the segments and the remaining end of one segment anchored to the world space. |
| Muscle | *ReluMuscle* | An actuator that produces forces according to a linear piece-wise function of activation. The lower bound of force production is 0. |
|  | *RigidTendonHillMuscle* | A Hill-type muscle according to the formalization in **Kistemaker et al., 2010**, adjusted for rigid-tendon dynamics. |
|  | *RidigTendonHillMuscleThelen* | A Hill-type muscle according to the formalization in **Thelen, 2003**, adjusted for rigid-tendon dynamics. |
|  | *CompliantTendonHillMuscle* | A Hill-type muscle according to the formalization in **Kistemaker et al., 2010**. |
| Effector | *ReluPointMass24* | A planar (2D) *PointMass* with four *ReluMuscle* actuators. |
|  | *RigidTendonArm26* | A *TwoJointArm* with six *RigidTendonHillMuscle* actuators. |
|  | *CompliantTendonArm26* | A *TwoJointArm* with six *CompliantTendonArm26* actuators. |
| Environment | *CentreOutReach* | A centre-out reaching task. |
|  | *DelayedReach* | A reaching task where movement initiation is signified by the appearance of a 'go' cue. |
|  | *DelayedMultiReach* | A reaching task where movement initiation is signified by the appearance of a 'go' cue, and several targets appear in sequence for each trial. |
| Policy | *PolicyGRU* | An RNN network comprising a user-defined number of layers containing a user-defined number of GRUs. |

from the *Environment*, *Effector*, *Skeleton*, and *Muscle* objects. This allows the user to monitor the true state of the MotorNet model at each timestep.

## Object structure

The classes presented above rely on each other to function correctly. Consequently, they must be declared in a sensible order, so that each object instance retains as attribute the object instances on which they rely. This leads to a hierarchical class structure, where each instance lives in the computer memory in a nested fashion with other instances, as laid out in *Figure 3b*. Note that this does not mean that each class is a subclass of the class that contains it, but that each contained class is saved as an attribute of the container class. The outermost class is an *Environment* object, which itself is a subclass of *gymnasium*'s *Env* class. The *Environment*, *Effector*, *Skeleton*, and *Muscle* objects are also *torch.nn.Module* subclasses. The *Policy* objects are distinct, in that they do not hold any other MotorNet object as attribute. This independence allows users to create their own neural networks without having to rely on MotorNet's *Policy* object, which allows for more freedom for the user to design any policy that PyTorch can create.

## Biomechanical properties of the effector

The modular structure detailed above allows MotorNet to flexibly compute detailed biomechanical properties of *Effector* objects, such as arbitrary muscle paths (**Nijhof and Kouwenhoven, 2000**), geometry-dependent moment arms (**Murray et al., 1995**; **Sherman et al., 2013**), non-linear muscle activations, and passive force production from muscle stretch (**Cheng, 2000**; **Millard et al., 2013**; **Thelen, 2003**). This enables training ANNs on motor tasks whose dynamics are highly non-linear and close to biological reality. In this section, we illustrate some of these biomechanical properties implemented by MotorNet effectors using specific examples. These properties are well characterized in the biology and are often implemented in realistic biomechanical simulation software.

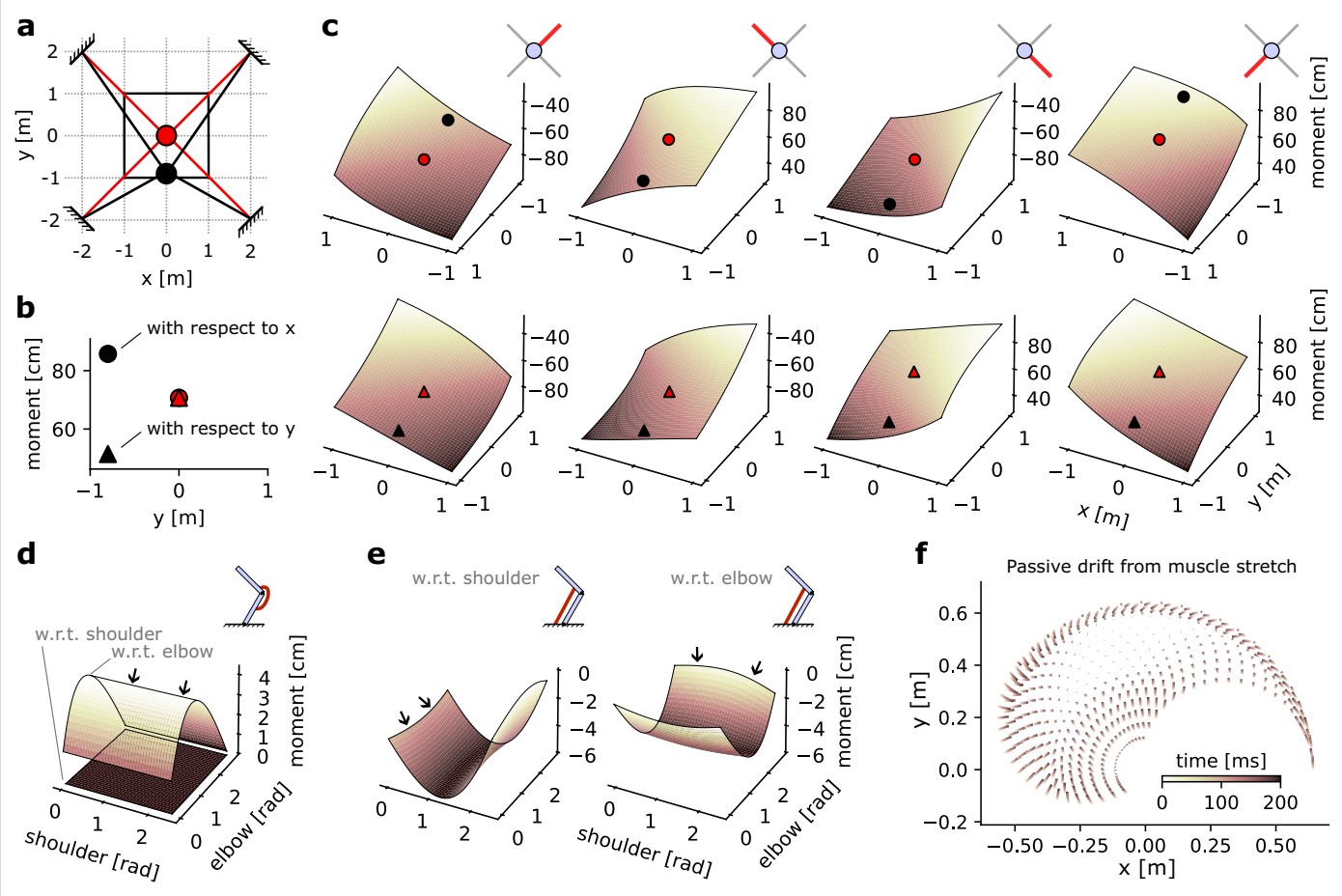

**Figure 4.** Geometrical properties of an Effector object. (**a**) Schematic of a point-mass in two positional configurations within a square workspace. The point-mass Skeleton was linked to four muscles in a 'X' configuration. (**b**) Moment arm values for the lower left muscle for each of the positional configurations represented in (**a**), with respect to $x$ and $y$. (**c**) Complete moment arm function over the position space for each muscle (columns) and with respect to each degrees of freedom (DoF). The upper and lower rows indicate the moment arm with respect to the $x$ and $y$ positions, respectively. (**d**) Moment arms of a mono-articular extensor muscle on an arm26. (**e**) Moment arms of a bi-articular flexor muscle on an arm26. (**f**) Passive drift in endpoint position of an arm26 similar to *Figure 1c* due to passive force developed by overstretch Hill-type muscles.

### Assessing moment arms with a simple point-mass effector

The geometrical path – fixation body(s) and fixation point(s) on that body – of each *Muscle* object can be declared by the user, allowing for arbitrary linkage between muscles and bones (see Methods section 'Biomechanical properties of the effector', **Nijhof and Kouwenhoven, 2000**). Using geometric first principles (**Sherman et al., 2013**), the *Effector* object can then calculate the moment arm of forces produced, which is defined for each muscle as the change in value of the DoF of the skeleton for a given change in the muscle's length (**Murray et al., 1995**; **Sherman et al., 2013**). In lay terms, this is the capacity of a muscle to produce a torque on a joint based on the muscle's pulling angle on the bones forming that joint. The relationship between pull angle and torque can intuitively be understood using a door as an example: it is easier to push a door when pushing with an angle orthogonal to that door than in a near-parallel angle to that door.

Moment arms generally vary depending on the positional configuration of the *Effector*. To illustrate this, let us consider a simple case of a point-mass skeleton (one fixation body) with four muscles attached to it in a 'X' configuration (*Figure 4a*). When the point-mass is positioned in the centre of the workspace space (red position in *Figure 4a, b*), any muscle pulling will change the position of the point-mass equally in the $x$ dimension and in the $y$ dimension. Note that $x$ and $y$ are the DoFs of the point-mass skeleton since they do not have hinge joints. In contrast, if the point-mass is positioned below the central position ($x = 0, y = -0.9$; black position in *Figure 4a*), a pull from for example, the

lower left muscle will produce a greater change in the $x$ dimension than in the $y$ dimension because of the different muscle alignment (*Figure 4b*).

The moment arm can then be calculated for all possible positions in the workspace, as represented by the solid black square in *Figure 4a*. This can be done for each of the four muscles, and each of the two DoFs, resulting in eight moment arms (*Figure 4c*). We can see that each moment arm forms a slightly bent hyperplane. Importantly, for each hyperplane the diagonal with constant moment arm lines up with the path formed by the muscle when the point-mass is at the centre of the workspace. For instance, the moment arm of the upper right muscle is identical when the point-mass is in position $(x = 1, y = 1)$ and in position $(-1, -1)$. This is true both with respect to the $x$ DoF (*Figure 4c*, upper row, leftmost axis) and with respect to the $y$ DoF (*Figure 4c*, lower row, leftmost axis). Note also that muscles whose shortening leads to an increase in the DoF considered – or inversely whose lengthening leads to a decrease in the DoF – express negative moment arms. For instance, a shortening of the lower right muscle would lead to an increase in the $x$ DoF and a decrease in the $y$ DoF. Or more plainly, a pull from the lower right muscle would bring the point-mass closer to the lower right corner of the workspace. This leads to the negative moment arm of that muscle with respect to $x$ (*Figure 4c*, upper row) and positive moment arm with respect to $y$ (lower row).

## Moment arms with a two-joint arm

To consider a more complex effector, we assessed the moment arm of two muscles wrapping around a two-joint arm skeleton. We first assessed a mono-articular muscle, that is, a muscle that spans only one joint – here, the elbow. As expected, the moment arm of that muscle with respect to the shoulder joint is always null (black arrows, *Figure 4d*) regardless of the joint configuration since the muscle does not span that joint. In contrast, the moment arm with respect to the elbow joint varies as the elbow joint angle changes. Finally, as expected from an extensor muscle, the moment arm is positive, indicating that the elbow angle would decrease as the muscle shortens.

In comparison, a bi-articular muscle's moment arm is non-zero with respect to both joints (*Figure 4e*). This also leads the moment arms with respect to each joint to show a small interaction as the other joint's angle changes, as indicated by a slight 'bend' in the hyperplane (black arrows, *Figure 4e*). Finally, as expected for a bi-articular flexor muscle, the moment arms are negative with respect to both joints, indicating that muscle shortening would result in an increase in joint angle.

## Passive drift with Hill-type muscles

Finally, we assessed the positional drift induced by passive forces of Hill-type muscle models (*Millard et al., 2013*; *Thelen, 2003*) in an *arm26* effector model. We initialized the model's starting position at fixed intervals across the range of possible joint angles, resulting in a grid of 21-by-21 possible starts. We then simulated the effector with null inputs for 200 ms and plotted the drift in the arm's endpoint position from its original position. Because the model received no input, all forces produced are due to the passive component of the Hill-type muscles, which occurs when the muscle is stretched beyond its slack length (*Cheng, 2000*; *Millard et al., 2013*; *Thelen, 2003*). We can see that drift is negligible at the centre of the joint space but starts to increase towards the edge (*Figure 4f*), indicating that the associated joint configurations lead to overstretched muscle lengths and resulting in passive force production. Note that since this phenomenon is dependent on the slack length value of each muscle, which is user-defined, the presence of passive drift is dependent on the user's modelling choices.

## Training ANNs to produce naturalistic behaviour

Now that we can implement biomechanically realistic effectors, we next assessed whether a policy ANN can learn a complex control policy to move those effectors using backpropagation (*Jordan and Rumelhart, 1992*; *Rumelhart et al., 1986*). A typical way to ensure the computation learnt by an ANN is functionally meaningful is to test its out-of-distribution generalization. To assess this, we trained a one-layer RNN with $n = 110$ GRUs controlling an *arm26* model to perform reaching movements in 0.8 s simulations using the following paradigm. Starting positions and targets were randomly drawn from a uniform distribution across the full joint space. Movements were to be delayed until the occurrence of a visual 'go' cue randomly drawn from a uniform distribution spanning the full simulation window. The appearance of the go-cue reached the RNN as input after a delay corresponding to the visual feedback delay, which was set at $\Delta_v = 50$ ms (*Figure 2*; *Dimitriou et al., 2013*; *Pruszynski*

*et al., 2010*). In half of trials, no go-cue was provided (catch trial), in which case the task effectively reduced to a postural control task. A 100-ms endpoint mechanical perturbation, whose orientation, magnitude, and time were also randomly drawn occurred in half of trials, independently of whether the trial was a catch trial or not. Importantly, the perturbation magnitude was drawn from a uniform distribution ranging between 0 and 4 N. If the perturbation occurred during a catch trial, the distribution ranged between 0 and 8 N. Therefore, the training protocol used for this task largely differed from section 'Training an ANN to perform a centre-out reaching task against a curl field' in that the networks are exposed to a wide range of mechanical perturbations with varying characteristics.

The network was trained using the following loss:

$$L = \frac{\sum_{t=1}^{T} L_t}{T} + \lambda \left\| W \right\|_2$$

$$L_t = \alpha\, L_t^p + \beta L_t^m + \gamma\, L_t^h$$

$$L_t^p = \begin{cases} 0, & \left\| x_t - x_t^* \right\|_2 < r \\ \left\| x_t - x_t^* \right\|_1, & else \end{cases}$$

$$L_t^m = \left( u_t^\top \frac{f}{\left\| f \right\|_2^2} \right)^2$$

$$L_t^h = \frac{h_t^\top h_t}{n} + \kappa \frac{\dot{h}_t^\top \dot{h}_t}{n} \tag{1}$$

With $L$ the global loss including a kernel regularization term with penalty coefficient $\lambda = 10e^{-6}$, and $W$ the kernel weight matrix of the RNN's hidden layer. The operators $\left\| \cdot \right\|_1$ and $\left\| \cdot \right\|_2$ indicate the L1 and L2 vector norm, respectively. $L_t$ is the instantaneous loss at time $t$, with coefficients $\alpha = 2, \beta = 5, \gamma = 0.1$. $L_t^p$ is the positional penalty at time $t$, with $x_t, x_t^*$ the position and desired position (target) vector, respectively, and $r = 0.01$ the target radius. $L_t^m$ is the muscle activation penalty at time $t$, with $u_t, f$ two vectors representing muscle activations at time $t$ and maximum isometric force, respectively. Finally, $L_t^h$ is the network hidden activity penalty at time $t$, with $h_t$ the $n$-elements vector of GRU hidden activity, $\dot{h}_t$ its time derivative, and $\kappa = 0.05$. While superficially this loss appears complex, a direct relationship to biology can be drawn for all terms. Essentially, this loss enforces the control policy to be learned using a simple, straightforward rule (get to the target), while promoting low metabolic cost from network input connectivity (cost on kernel norm), from the muscles (cost on activation, scaled by muscle strength), and from network activity (cost on hidden activity and its derivative to discourage oscillatory regimes).

Behavioural performance on a training set can be seen in *Figure 5a*, with trials with a large perturbation (> 3N) highlighted in blue. This illustrates the rich variability of the training set, encouraging the RNN to learn computationally potent and generalizable solutions to the control problem given the sensorimotor feedback provided (*Figure 2*). Despite this variability, the loss value decreased smoothly (*Figure 5b*).

We tested the model's behavioural output in 0.8 s simulations with a centre-out reaching task. Eight targets were positioned in 45 degrees increments and 10 cm away from a starting position corresponding to a shoulder and elbow angle of 45 and 90 degrees, respectively (*Figure 5c*). The RNN reached to each of these targets following a visual go-cue at 100 ms. 70 ms after the 'go' cue was 'perceived' (i.e., 70 ms plus the visual feedback delay), a mechanical perturbation was applied at the arm's endpoint and orthogonally to the reaching direction. This perturbation could be either within-distribution (±3 N) or out-of-distribution (±6 N) or null (no perturbation). In all cases, the RNN could correct for the mechanical perturbation, reach to the target, and stabilize (*Figure 5c*).

Next, we tested the RNN in a postural control task, where it had to bring the arm's endpoint back to the target following a mechanical perturbation (*Pruszynski et al., 2014*). No go-cue was provided. We applied perturbations in either of the four cardinal directions (0°, 90°, 180°, and 270°) at 170 ms plus visual delay after the trial started. Again, the set of perturbations for testing outputs included within-distribution magnitudes (±6 N) and out-of-distribution magnitudes (±12 N). In all cases, the RNN could integrate the sensorimotor information to bring the arm's endpoint back into the target (*Figure 5d*). Interestingly, in some cases this led to an oscillatory trajectory (e.g., for a rightward +12 N

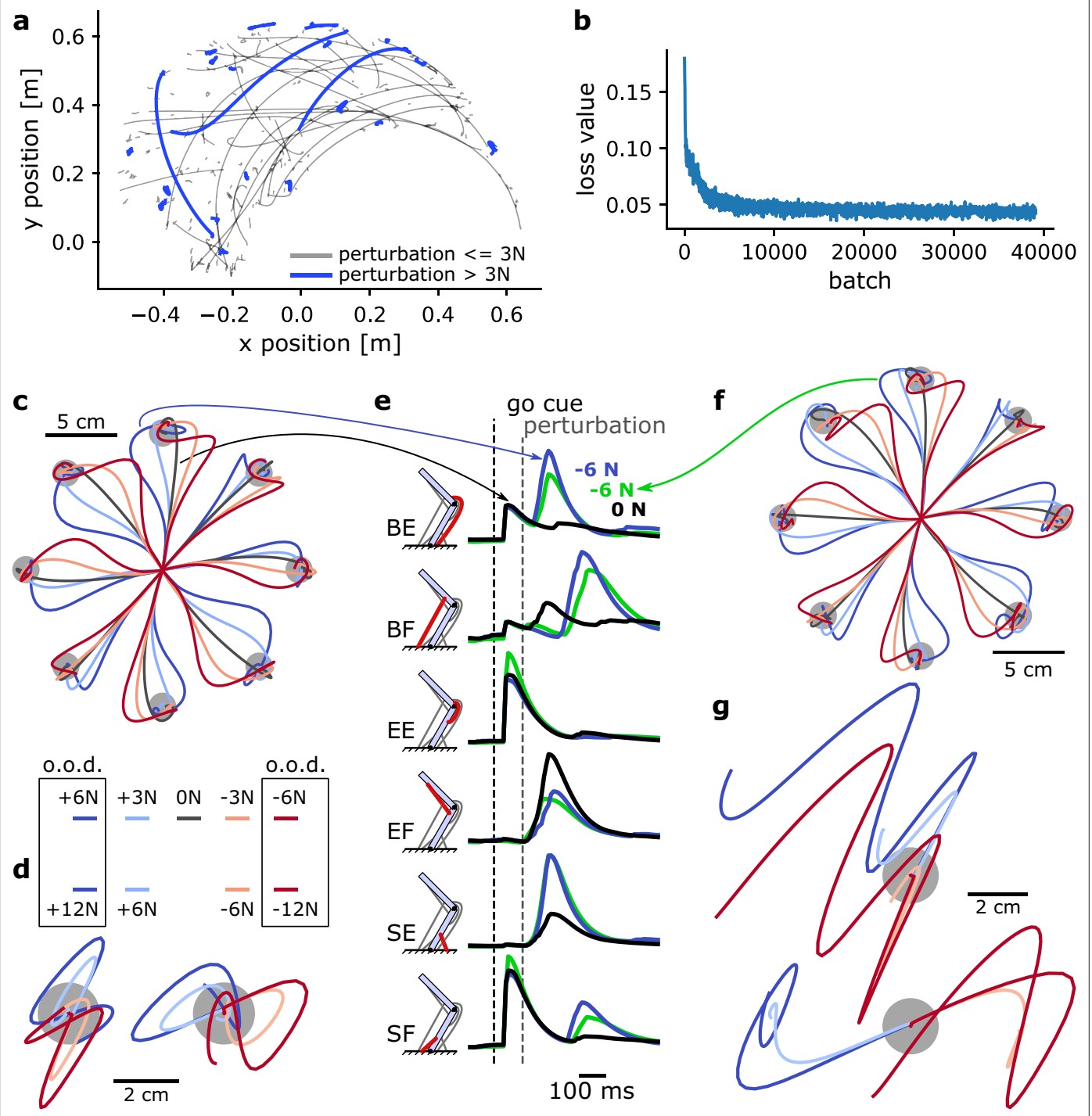

**Figure 5.** A MotorNet model can learn a control policy that generalizes to out-of-distribution perturbations. (**a**) Example endpoint trajectories produced by a network after training. (**b**) Loss function over training iterations, with a batch size of 1024. (**c**) Trajectories in a centre-out reaching task with mechanical perturbations applied at the arm's endpoint 120 ms after the 'go' cue. The perturbations were orthogonal to the reaching axis passing from the starting position to the target. o.o.d.: out-of-distribution. (**d**) Same as (**c**) for a postural control task. In this task, the network was not provided with a target and therefore only had to remain in the starting position against the perturbations. Mechanical perturbations were in the vertical (left) or horizontal (right) axis. (**e**) Muscle activation over time for two trajectories in (**c**) (black and blue lines) and a trajectory in (**g**) (green line). BE: bi-articular extensor; BF: bi-articular flexor; EE: elbow extensor; EF: elbow flexor; SE: shoulder extensor; SF: shoulder flexor. (**f**) Reaching task as in (**c**) for a network never exposed to mechanical perturbations during training. (**g**) Postural task as in (**d**) for the same network as in (**f**). Perturbations were in the vertical (top) or horizontal (bottom) axis.

perturbation, *Figure 5d*), indicating that perturbations beyond a given magnitude remain increasingly challenging to control for.

Finally, we compared muscle activations for an upward reach with no perturbation to that of an identical reach with a −6 N perturbation (*Figure 5e*). We can see that muscle activations are similar before the occurrence of the perturbation, and remain similar immediately after, indicating a time delay in the response. The fastest responses occurred for the bi-articular muscles and the shoulder extensor muscle. Other muscles, particularly the shoulder flexor, showed very delayed or non-existent changes in muscle activation. This illustrates that the RNN's response to a perturbation is not a mere stimulus-driven reactive response, but an integrated response that can delay or withhold the production of counteracting forces if necessary. Note that for the non-perturbed movement (black line in *Figure 5e*), we can observe the canonical tri-phasic muscle activation pattern reported in empirical studies (*Wierzbicka et al., 1986*).

To assess how the existence of sensorimotor feedback impacted the control policy acquired by the policy network, we trained a second, identical network to perform the same task but with no mechanical perturbation during training (perturbation-free). Interestingly, following the same amount of training, the model with a perturbation-free network can handle perturbations during reaching relatively well, even up to ±6 N (*Figure 5f*). We can compare muscle activations for an upward reach with a −6 N perturbation to that of the same movement in the network trained with perturbations (*Figure 5f*, green versus blue lines). Even though kinematics appeared superficially similar (*Figure 5c, f*), this comparison shows that muscle activations tend to differ in response to a perturbation (*Figure 5e*), suggesting that the perturbation-free network might learn a slightly different control policy. Testing the perturbation-free network on the postural task shown in *Figure 5d* emphasizes this difference (*Figure 5g*). The perturbation-free network is much less capable of stabilizing against the forces than its perturbation-trained counterpart.

Therefore, even though the mere existence of a sensorimotor feedback input can help handle simple perturbations (*Figure 5f*), exposing the model to perturbations during training does provide the network with additional information to learn a more robust control policy. Overall, these simulations show that MotorNet can train ANNs to reliably find a control policy for the effector. Importantly, the resulting networks learn generalizable control policies that integrate sensorimotor feedback into its computation. This also illustrates the importance of the training procedure to which the network is exposed to produce these control policies (*Driscoll et al., 2022*).

## Effector geometry defines preference distribution of firing rates: a replication study

Finally, to assess MotorNet's capacity to replicate established results in the literature, we sought to reproduce key observations from *Lillicrap and Scott, 2013*. In that study, the authors show that training an RNN to perform a simple centre-out reaching task using an arm model similar to the arm26 in *Figure 1a* results in the RNN neurons displaying a preferential movement direction (PMD) where they are more likely to fire. The distribution of PMDs was asymmetrical, with a greater proportion of neurons firing for reaches around 135 and 325 degrees, matching empirical observations from non-human primate electrophysiological recordings in the primary motor cortex (*Scott et al., 2001*). Next, they showed that this asymmetrical representation of PMDs during reaching movements did not occur when RNNs were trained to control an effector that lacked the geometrical properties of an arm such as illustrated in *Figure 4c–e* and section 'Training an ANN to perform a centre-out reaching task against a curl field'. Specifically, they compared the PMD distribution of RNN neurons controlling a point-mass (no geometry) against that of an arm26 (geometry present).

We sought to reproduce the two results outlined above. First, we trained an RNN composed of 90 GRUs in a single layer to control for an arm26 (*Figure 6a* see Methods section 'Effector geometry defines preference distribution of firing rates: a replication study'). Because our RNN employs GRUs instead of a multi-layer perceptron, 90 units were sufficient to efficiently train the network to perform the task, as opposed to up to 1000 perceptron nodes in the original study. We also increased the number of targets from 8 to 24 to obtain a finer resolution over movement direction in our analyses (*Figure 6b*).

Following training, we first ensured that muscle activation patterns in the arm26 effector were like those reported in the original study (*Figure 6d*). Regarding network activity, we observed a great

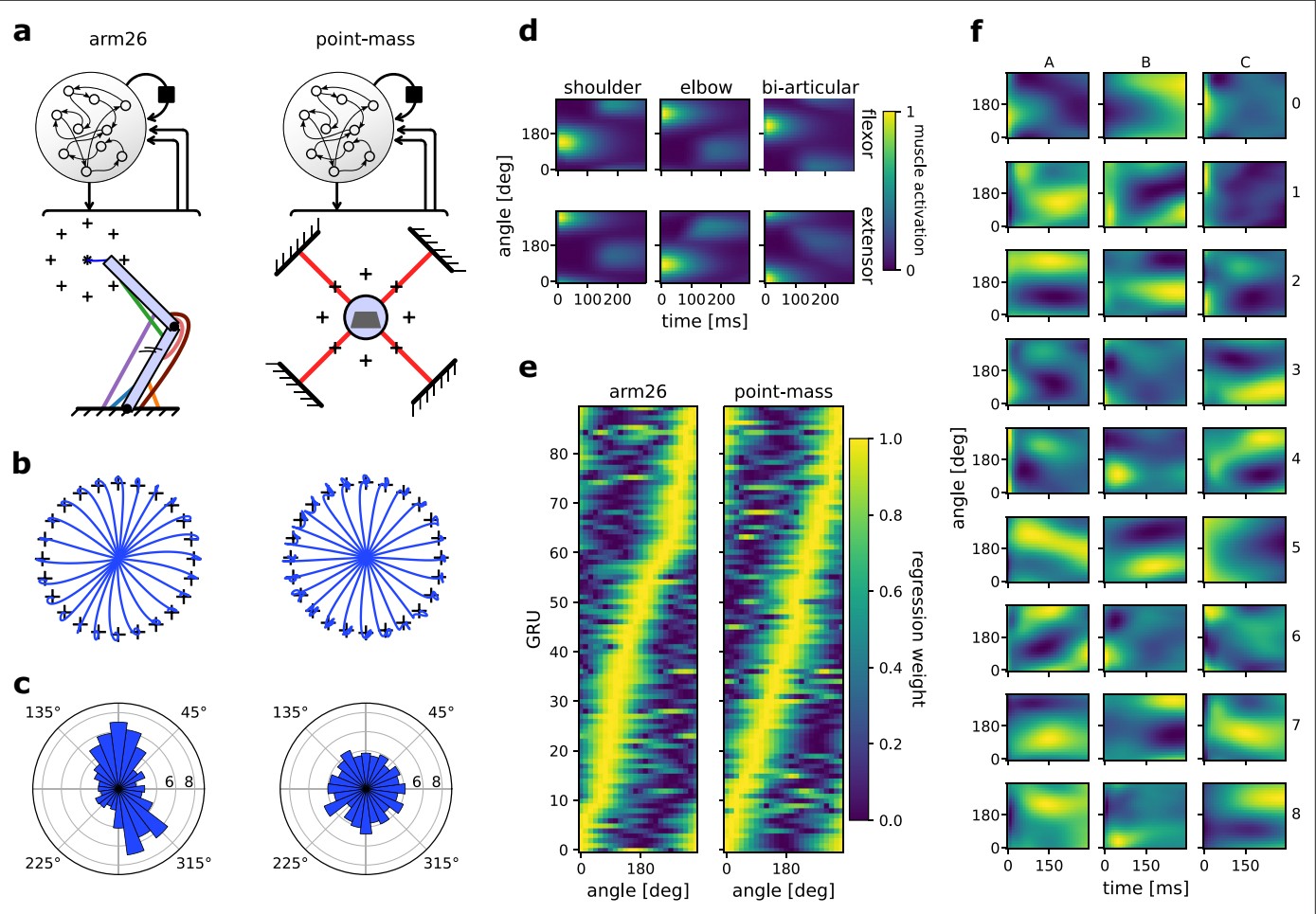

**Figure 6.** The distribution of preferential movement direction (PMD) tuning is sensitive to the geometry of the effector. (**a**) Schematic of the two models compared. The recurrent neural networks (RNNs) and their architecture were identical, but the effector differed, with one RNN controlling a two-joint arm26 (left) and the other controlling a point-mass (right). (**b**) Centre-out reaching trajectories to 24 targets for the arm26 (left) and point-mass (right) model. (**c**) Distribution of PMDs for the arm26 (left) and point-mass (right) model. The PMDs were determined by regression of each gated recurrent unit's (GRU) hidden activity averaged over time against reach angle (see Methods for details). (**d**) Normalized muscle activations across reaching angles and for the 300 ms following the 'go' cue for the arm26 model. (**e**) Normalized $\beta$ coefficients of the regression models used for (**c**). The GRUs were ordered according to the angle of their maximum $\beta$ value. Note that the 'ridge' of maximum $\beta$ yields roughly a straight line for the point-mass model, while it yields a crooked line for the arm26, indicative of a representation bias. (**f**) Hidden activity over time and across reaching angles for a random sample of GRUs in the arm26 model.

variety of activation patterns over movement direction (*Figure 6f*). Some GRUs showed a preference for timing (e.g., neuron A4, C5), while others showed a strong preference for reaching direction that was sustained over time (neuron C3, A2). Finally, most neurons showed a mixed preference for encoding time and reaching direction (neuron C8, A8). This heterogeneous set of responses matches empirical observations in non-human primate primary motor cortex recordings (*Churchland and Shenoy, 2007*; *Michaels et al., 2016*) and replicate similar visualizations from previously published work (*Fortunato et al., 2024*; *Lillicrap and Scott, 2013*; *Safaie et al., 2023*).

We then assessed each GRU's PMD using linear regression (see methods) and sorted them based on their PMD before plotting the tuning curve of each neuron. The resulting colormap (*Figure 6e*, left panel) yields a 'ridge' of maximal activity whose peak varies across reach angle, forming a crooked line, illustrating a representational bias. This crooked ridge line was not observed in an RNN trained to control for a point-mass effector instead using an identical training procedure and analysis (*Figure 6e*, right panel). We replicated this procedure with seven more RNNs for each model, resulting in a total of eight RNNs trained on an arm26 and eight RNNs trained on a point-mass. We determined each GRU's PMD and averaged the resulting polar histogram across each RNN (*Figure 6c*). The same bias

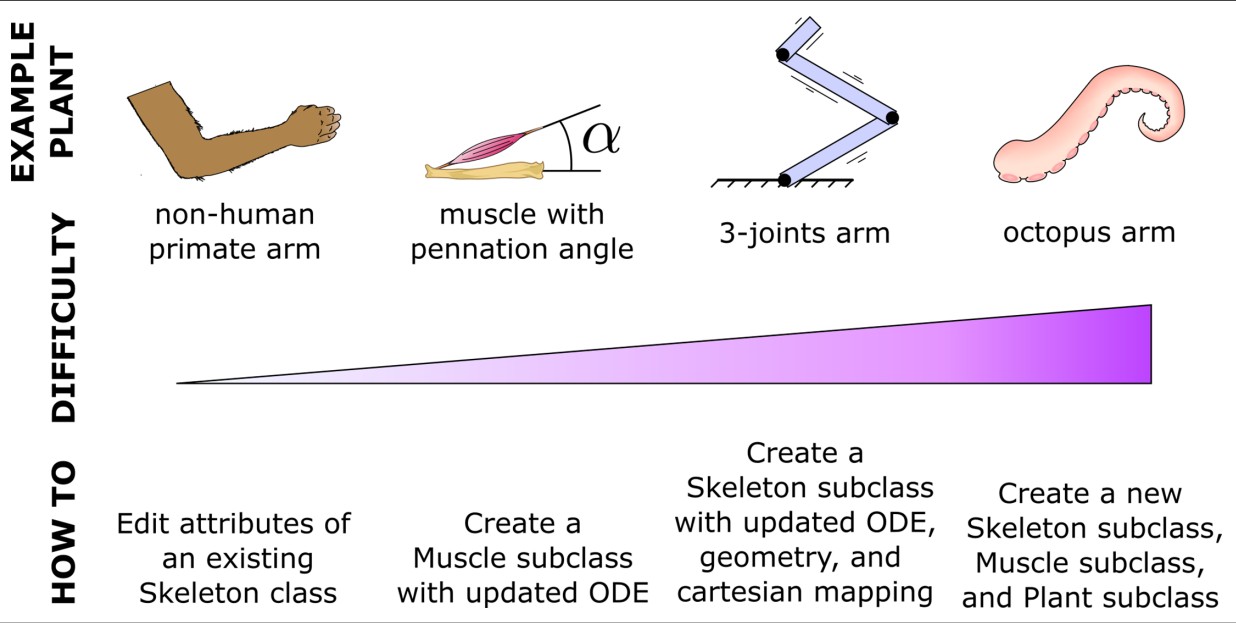

**Figure 7.** MotorNet is expandable. MotorNet allows for new features to be implemented through subclassing.

was reproduced invariably for the RNNs controlling an arm26 effector (*Figure 4d, e*, *6a*), while it failed to arise for those controlling a point-mass (*Figure 4a–c*, *6a*). Therefore, these results mimic the observations made in the original study (*Lillicrap and Scott, 2013*), specifically, that RNNs controlling an effector with no arm-like geometrical properties will not result in the biased PMD representation during reaching movements commonly observed in non-human primate electrophysiological studies (*Scott et al., 2001*; *Scott and Kalaska, 1997*).

## Discussion

### Iterating quickly through the model development cycle

In the field of ML, an established best practice is to iterate quickly around a cycle of (1) formulating an idea, (2) implementing that idea in functionally efficient code, and (3) testing the idea through running the simulations. The results of the simulations can then be leveraged to adjust the idea, thus closing the loop, and enabling iterative refinement of a model. This [idea → code → test → idea] cycle is reminiscent of the [hypothesis → design task → test → hypothesis] cycle in empirical work, also known as the hypothetico-deductive method. An important practice in ML is to ensure that one iteration of that cycle is quick enough, because producing an efficient model may require many such iterations. Based on this framework, a way to view MotorNet is to improve iteration speed through this cycle. The modular architecture of MotorNet enables users to alter specific aspects of the model while keeping everything else identical. Therefore, user capacity to proceed through the 'implementation' step is enhanced.

### Advantages

#### Expandability

MotorNet naturally allows users to create and tune objects to fit individual requirements. This makes the toolbox easily expandable to add novel models that are not pre-built in the original distribution. This flexibility will likely vary depending on the goal (*Figure 7*). Some extensions only require adjusting parameter values of existing object classes, such as editing the Arm26 *Skeleton* class to match the arm of a non-human primate. Other extensions will require subclassing, such as creating an *Effector* for an eyeball, which might require special geometric properties building on the point-mass *Skeleton* object (*Table 1*). Conversely, effectors that stray away from typical vertebrate effectors will likely prove more challenging, such as an octopus arm, because they do not rely on bones. Importantly, while all these

extensions vary in the difficulty of their implementation, each has the capacity to fit and work harmoniously within the framework of the MotorNet architecture.

## Open source

Typically, when motor control researchers want to create canonical models, they must implement their own version of said model based on methodological descriptions of previously published scientific articles. However, because MotorNet is open source, individual contributions can easily be shared online for the benefit of others. For instance, if a researcher creates a *Muscle* class with a parametrizable pennation angle (***Millard et al., 2013***; ***Thelen, 2003***), future researchers and team will not have to re-create their own implementation of the same object anymore. This also allows more dynamical peer-checking, avoiding dissemination of errors and improving consistency of model implementations. In other words, MotorNet will be able to benefit from community-driven incremental work through open-source practices.

## Innovation scalability

For the past several years, ML has been standing out as one of the most dynamic research fields, achieving breakthroughs and successfully scaling innovative work towards solving everyday problems. It would be challenging for MotorNet to keep up with the pace of ML innovation to provide users with implementations of the latest architectures and algorithms. Rather, we rely on PyTorch to build policies. This ensures that any innovation in model design quickly finds its way to a viable MotorNet implementation, because PyTorch capabilities allow for fast adaptation aligned with progress in ML. Generally, MotorNet is built with the following logic in mind: anything PyTorch can build, MotorNet should be able to use as a policy.

## *gymnasium*-compliant interfacing

The MotorNet *Environment* class is a subclass of *gymnasium*'s *Env* base class and abides by its associated API (***Chinnaiya et al., 2023***). Consequently, MotorNet environments are by design compatible with any Python toolbox that works with *gymnasium*, which is a standard and popular interfacing toolbox to link reinforcement learning agents with environments. It is very well documented and widely used, which will ensure that users who wish to employ reinforcement learning to control MotorNet environments will be able to do so relatively effortlessly.

## Limitations
### Collision physics

Typical biomechanical software distributions implement some form of collision physics in their physics engine (***Delp et al., 2007***; ***Seth et al., 2018***; ***Todorov et al., 2012***). This is not the case for MotorNet.

### Complex biomechanical features

Some biomechanical software distributions such as OpenSim propose a large array of joint types such as hinge joints or rotational joints, and complex muscle paths such as wrap points that trigger only when the muscle collides with them (***Delp et al., 2007***; ***Seth et al., 2010***; ***Seth et al., 2018***). While these features increase the realism of a biomechanical model, MotorNet does not yet implement these types of features. In practice, this constrains what types of effectors MotorNet can realistically implement and adding some of these features is under consideration.

## Future considerations

As an open-source, freely available Python toolbox, MotorNet is subject to change over time. Some of the limitations outlined above are considered as future routes for improvement. Additionally, we hope that individual contributions will help refine and extend the capabilities of the toolbox as well. In this section, we outline prospective improvements for implementation and release in the main distribution.

**Table 2.** Skeleton parameters for the arm26 model, taken from *Nijhof and Kouwenhoven, 2000*.

The skeleton was actuated by six rigid-tendon versions of Hill-type muscle actuators: a shoulder flexor, a shoulder extensor, an elbow flexor, an elbow extensor, a bi-articular flexor, and a bi-articular extensor. Their parameter values are defined in *Table 3*.

| Parameter | Upper arm | Forearm |
|---|---|---|
| Mass (kg) | 1.82 | 1.43 |
| Centre of gravity (m) | 0.135 | 0.165 |
| Inertia (kg·m²) | 0.051 | 0.057 |
| Length (m) | 0.309 | 0.333 |

## Spinal Compartment

It is becoming increasingly evident that spinal contribution plays a prominent role in motor control beyond the typically considered spinal reflex (*Reschechtko and Pruszynski, 2020*; *Weiler et al., 2019*). One may consider that supraspinal control interacts with spinal contribution to define a motor control policy (*Loeb, 2021*). Within MotorNet, this suggests that a policy's latent dynamics will be significantly impacted by the presence of a spinal compartment. Consequently, it may be worthwhile to implement one such spinal compartment to explore the consequences of such biological design, especially with regard to upstream computation (*Cisek, 2019*). Importantly, this compartment may be designed as a module within the controller downstream from the ANN, that instantiates arbitrarily detailed computation according to empirical studies. This could include processing of top–down and/ or bottom–up information as a movement unfolds (*Reschechtko and Pruszynski, 2020*) with appropriate time delays in place. Ultimately, the scientific question at hand will dictate the complexity desiderata for the spinal compartment implementation. Interestingly, this two-module design within the controller leans towards the more general concept of modular architectures, which can be powerful for understanding multi-region interactions within the central nervous system (*Michaels et al., 2020*).

## Modular policies

A deeply established idea in neuroscience is that distinct regions will perform different computations, and thus that a complex system may not be considered as a uniform, fully connected network (*Abbott and Svoboda, 2020*; *Keeley et al., 2020*; *Pesaran et al., 2021*; *Semedo et al., 2020*). This is also true for the motor control system, where using a modular network architecture with controlled communication between each module has been shown to have more explanatory power than a non-modular system (*Michaels et al., 2020*). Therefore, a potential development for MotorNet is to include a model class with a modular architecture to study how cross-region networks work to enable neural control of the body.

## Muscle models

Most published work in motor control relies either on Hill-type muscle models (*Bhushan and Shadmehr, 1999*; *Kistemaker et al., 2006*; *Kistemaker et al., 2010*; *Nijhof and Kouwenhoven, 2000*) or

**Table 3.** Parameters for the Hill-type muscle actuators used in the arm26, taken from *Kistemaker et al., 2010*.

| Muscle | Maximum isometric force (N) | Tendon length (m) | Optimal muscle length (m) |
|---|---|---|---|
| Shoulder flexor | 838 | 0.039 | 0.134 |
| Shoulder extensor | 1207 | 0.066 | 0.140 |
| Elbow flexor | 1422 | 0.0172 | 0.092 |
| Elbow extensor | 1549 | 0.187 | 0.093 |
| Bi-articular flexor | 414 | 0.204 | 0.137 |
| Bi-articular extensor | 603 | 0.217 | 0.127 |

**Table 4.** Parameters used to compute moment arms in the arm26 models with moment arm approximation, taken from *Kistemaker et al., 2010*.

| Muscle | $a_0$ | $a_{1e}$ | $a_{1s}$ | $a_{2e}$ |
|---|---|---|---|---|
| Shoulder flexor | 0.151 | 0 | −0.03 | 0 |
| Shoulder extensor | 0.2322 | 0 | 0.03 | 0 |
| Elbow flexor | 0.2859 | −0.014 | 0 | −4.0e−3 |
| Elbow extensor | 0.2355 | 0.025 | 0 | −2.2e−3 |
| Bi-articular flexor | 0.3329 | −0.016 | −0.3 | −5.7e−3 |
| Bi-articular extensor | 0.2989 | 0.03 | 0.03 | −3.2e−3 |

direct torque actuators (*Lillicrap and Scott, 2013*) similar to the ReLu muscle that MotorNet provides. However, despite its popularity, even the more-detailed Hill-type muscle remains a phenomenological model of real muscle behaviour, which can easily show its limits when trying to understand how the brain controls movement (*Blum et al., 2020*). Alternative muscle model formalizations exist, such as the Distribution-Moment muscle model (*Zahalak, 1981*), which may be worth implementing within MotorNet as well.

## Methods
### General modelling design
This section describes modelling elements that were used for several models in this study. For all models, the timestep size was 0.01 s, and a proprioceptive delay of $\Delta_p = 20$ ms and visual delay of $\Delta_v = 50$ ms were used (*Figure 2*). Effectors were actuated using numerical integration with the Euler method.

### Arm26 model
The arm26 model used in this study is available online on the open-source toolbox code under the *RigidTendonArm26 Effector* class. It is briefly described below for convenience.

The skeleton of the arm26 models are according to the formalization proposed in *Gomi and Kawato, 1997*, *equations 1, 3, 5–7*. Parameter values are as in *Table 2*.

The full formalization of the Hill-type muscles can be found in *Thelen, 2003*, *equations 1–7*, and with the parameter values used in that study. When different parameters were provided for young and old subjects, the values for young subjects were used (*Thelen, 2003*, *Table 1*). While in custom-made *Effector* objects the moment arms of each muscle are computed based on geometric first principles (*Figure 4d–f*; *Sherman et al., 2013*), in the *RigidTendonArm26* class the moment arms are approximated as described in *Kistemaker et al., 2010*, *equations A10–A12*, with parameters for this study defined in *Table 4*.

### Point-mass model
The point-mass model used in this study is available online on the open-source toolbox code under the *ReluPointMass24 Effector* class. It is briefly described below for convenience.

The point-mass had a mass of $m = 1$ kg. Its actuation followed an ordinary differential equation such that $x = f/m$ with $x, f$ the two-element cartesian acceleration vector at time $t$ and the two-element force vector applied at time $t$, respectively.

The forces were produced by four linear muscle actuators, whose formalization is available online on the open-source toolbox code under the *ReluMuscle* muscle class. Each muscle's force production $f$ is a linear piecewise function of its activation $a$, scaled by its maximum isometric force $f_{max} = 500$ N:

$$\mathrm{f}(a) = \begin{cases} 0, & a \leq 0 \\ f_{max}.a, & 0 < a < 1 \\ f_{max}, & a \geq 1 \end{cases}$$

The activation function was the same as for the Hill-type muscles used in the arm26 model, and can be found in *Thelen, 2003*, *equations 1 and 2*.

The four muscles were fixed to the point-mass in a 'X' configuration (*Figures 4a and 6a*) with the first fixation point for the upper right, lower right, lower left, and upper left muscle being, respectively $(x = 2, y = 2)$, $(2, -2)$, $(-2, -2)$, $(-2, 2)$. The second fixation point of each muscle was on the point-mass, therefore moving in general coordinates alongside the point-mass (*Figure 4a*).

## Policy network architecture

All policy networks used in this study consisted of one layer of GRUs with a sigmoid recurrent activation function and a *tanh* activation function. Kernel and recurrent weights were initialized using Glorot initialization (*Glorot and Bengio, 2010*) and orthogonal initialization (*Hu et al., 2020*), respectively. Biases were initialized at 0.

The GRU layer was fully connected to an output layer of perceptron nodes with a sigmoid activation function. The output layer contains one node per descending action signal, or equivalently one node per muscle. The output layer's kernel weights were initialized using a random normal distribution with a standard deviation of 0.003, and its bias was initialized at a constant value of −5. Because the output activation function is a sigmoid, this initial bias forces the output of the policy to be close to 0 at the start of initialization, ensuring a stable initialization state.

For all networks used in this study, a two-element vector of $(x, y)$ cartesian coordinates for the start position and target position were provided as input, alongside a go-cue, resulting in a five-element input vector. The go-cue was a 'step' signal whose value changed from 1 to 0 when the movement should be initiated.

## General training design

During training, the models reached from a starting position drawn from a random uniform distribution across the full joint space to a target position drawn from a random uniform distribution as well. The occurrence time of the go-cue was drawn from a random uniform distribution across the full simulation duration. In 50% of simulations, no go-cue was provided (i.e., a catch trial) to ensure the network learnt to wait for the go-cue and avoided any anticipatory activity. The desired position $x^*$ was set to be the start position until the go-cue was provided, at which point $x^*$ was defined as the target position. Note that the go-cue was treated as a visual signal. Therefore, while the desired position $x^*$ was updated immediately as the go-cue was provided (with no time delay), the network was informed of the go-cue occurrence via a change in the target position input and go-cue input only following the visual feedback delay $\Delta_v$. Depending on the models, additional training manipulations were also applied, as described in the sections below.

## **Centre-out reaches task against a curl field**

### Model

The effector type used to learn to reach against a curl field was an arm26 model as described in section 'Arm26 model'. The policy was as described in section 'Policy network architecture', with the GRU layer containing $n = 50$ units.

### Training

The model was trained according to the procedure in section 'General training design' with the loss described in *Equation 1*, using a kernel regularization $\lambda = 10e^{-6}$, coefficients $\alpha = 2, \beta = 5, \gamma = 0.1, \kappa = 0.05$, and target radius $r = 0.01$ m. The model was trained on 7680 batches with a batch size of 64, on simulations of 1 s.

The model was trained according to section 'General training design', except that the go-cue time was fixed at 100 ms from the start of the simulation. Following initial training, the model was then

tested against a null field and external forces to produce the 'naive' behaviour shown in *Figure 1b, c*. Following testing, training was then resumed, but employing the curl-field, fixed starting position, and set of eight targets used in testing. 50% simulations were still catch trials, as in the initial training session. This second training session lasted 768 batches with a batch size of 64. Finally, following this second training session, the model was tested again, to produce the 'adapted' behaviour of *Figure 1b*.

## Testing

The model was tested in 1 s simulations against a null field, and against external forces applied at the arm's endpoint that produced a counter-clockwise curl field:

$$f_t = b \begin{bmatrix} 0 & -1 \\ 1 & 0 \end{bmatrix} \dot{x}_t \tag{2}$$

With $\dot{x}_t$ the two-element cartesian velocity vector at time $t$, and $b = 8$ a scalar defining the strength of the curl field. In the null field, we have $b = 0$.

The testing procedure consisted of eight centre-out reaches from a fixed starting position at a shoulder and elbow angle of 45° and 90°, respectively, to eight target positions 10 cm away and distributed in increments of 45° around the starting position (*Figure 1b*). This set of simulations were repeated against a null field and against the curl field in *Equation 2*, resulting in a total of 16 reaches. For all testing simulations, the go-cue time was fixed at 100 ms from the start of the simulation and no catch trials were employed.

## Biomechanical properties of the effector

The point-mass model used was as described in section 'Point-mass model'. The arm26 model used was as described in section 'Arm26 model', except that the moment arms were not approximated based on the parameters of *Table 4*, but computed based on the geometry of the muscle paths

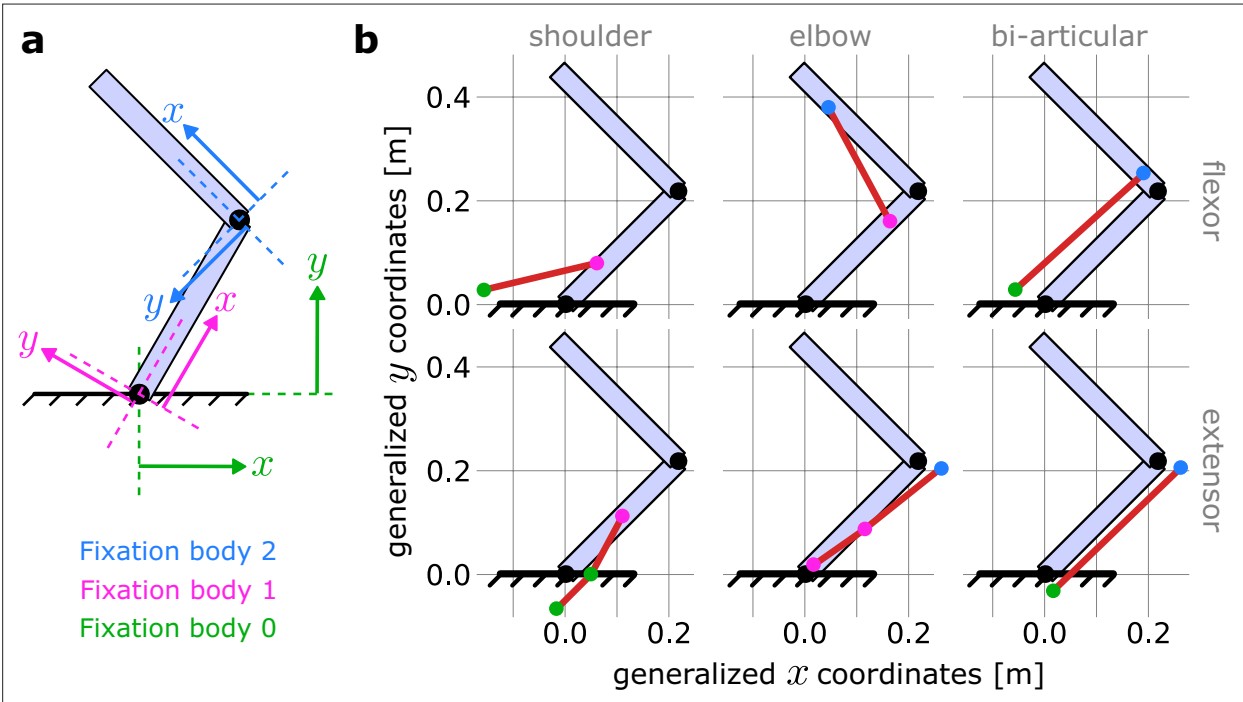

**Figure 8.** Coordinate frames for declaring muscle paths in MotorNet. (**a**) MotorNet handle muscle paths using coordinate frames relative to the bone on which a fixation point is. The world space is indexed as the fixation body '0' and its coordinate frame is the general coordinate system. (**b**) Schematic illustration of the muscle paths used for the arm26 model with no moment arm approximation described in section 'Biomechanical properties of the effector' and *Table 5*, for a shoulder and elbow angle of 45° and 90°, respectively.

**Table 5.** Muscle paths for the arm26 model with no moment arm approximation.

| Muscle | Fixation point | Fixation body | First coordinate $x$ (m) | Second coordinate $y$ (m) |
|--------|----------------|---------------|--------------------------|---------------------------|
|        | 1              | 0 (world)     | −0.15                    | 0.03                      |
| SF     | 2              | 1 (upper arm) | 0.094                    | 0.017                     |
|        | 1              | 0 (world)     | −0.013                   | −0.07                     |
|        | 2              | 0 (world)     | 0.05                     | 0                         |
| SE     | 3              | 1 (upper arm) | 0.153                    | 0                         |
|        | 1              | 1 (upper arm) | 0.23                     | 0.001                     |
| EF     | 2              | 2 (forearm)   | 0.231                    | 0.01                      |
|        | 1              | 1 (upper arm) | 0.03                     | 0                         |
|        | 2              | 1 (upper arm) | 0.138                    | −0.019                    |
| EE     | 3              | 2 (forearm)   | −0.04                    | −0.017                    |
|        | 1              | 0 (world)     | −0.052                   | 0.033                     |
| BF     | 2              | 2 (forearm)   | 0.044                    | 0.001                     |
|        | 1              | 0 (world)     | 0.02                     | −0.028                    |
| BE     | 2              | 2 (forearm)   | −0.04                    | −0.017                    |

(*Nijhof and Kouwenhoven, 2000*; *Seth et al., 2010*; *Sherman et al., 2013*). Accordingly, the muscle paths were manually declared by defining how many fixation points each muscle has, and on which bone and where on each bone each point fixes.

MotorNet handles declaration of these paths using a relative reference frame for each fixation point (*Seth et al., 2010*). Specifically, a fixation point on a bone will have two coordinates. The first coordinate defines how far along the bone the point is, from the bone's origin, for example, the shoulder for the upper arm (*Figure 8*). The second coordinate defines how far the point deviates from the bone orthogonally. If the fixation point is an anchor point, that is, it is not fixed on a bone but on the world space, then general coordinates $(x, y)$ are used (colour-coded in green in *Figure 8*). These anchor points are important to ensure that the effector can be actuated with respect to the environment. The full set of coordinates defining the model's muscle paths are indicated in *Table 5* and are derived from *Nijhof and Kouwenhoven, 2000*.

### Training ANNs to produce naturalistic behaviour
#### Model
The two models used to produce *Figure 5* were arm26 models as described in section 'Arm26 model'. For both models, the policy was as described in section 'Policy network architecture', with the GRU layer containing $n = 110$ units. In addition, excitation and GRU hidden activity noise were added, with values $\sigma_{\mathrm{u}} = 10e^{-3}, \sigma_{\mathrm{h}} = 10e^{-4}$, respectively.

#### Training
The models were trained with the loss described in *Equation 1*, using a kernel regularization $\lambda = 10e^{-6}$, coefficients $\alpha = 2$, $\beta = 5$, $\gamma = 0.1$, $\kappa = 0.05$, and target radius $r = 0.01$ cm. The model was trained on 27,000 batches of size 1024, on simulations of 800 ms.

In one of the two models, which we refer to as the 'perturbation-free' model, the training procedure was as described in section 'General training design'. In the second model, which we refer to as the 'perturbation-trained' model, a 100-ms endpoint mechanical perturbation was added to the training procedure. The perturbation occurred in 50% of trials, independently of whether the trial was a catch trial or not, and its orientation and time were randomly drawn as well. The magnitude of the perturbation was drawn from a uniform distribution ranging between 0 and 4 N. If the perturbation occurred during a catch trial, the distribution ranged between 0 and 8 N.

## Testing

Both the perturbation-trained and -free models were tested in 800 ms simulations in two distinct tasks, a centre-out reaching task and a postural task.

In the centre-out reaching task, eight targets were positioned in 45 degrees increments and 10 cm away from a starting position corresponding to a shoulder and elbow angle of 45° and 90°, respectively (*Figure 5c, g*). The visual go-cue was provided at 100 ms following the simulation start. 70 ms after the go-cue was 'perceived' (i.e., 70 ms plus the visual feedback delay $\Delta_v$), a mechanical perturbation was applied at the arm's endpoint and orthogonally to the reaching direction. This perturbation could be either within-distribution (±3 N) or out-of-distribution (±6 N) or null (no perturbation).

In the postural control task, no go-cue was provided, and the arm's endpoint was pushed away from the start position by the mechanical perturbation at 170 ms plus visual delay $\Delta_v$ after the simulation started. We applied perturbations in either of the four cardinal directions (0°, 90°, 180°, and 270°). Again, the set of perturbations for testing outputs included within-distribution magnitudes (±6 N) and out-of-distribution magnitudes (±12 N).

## Effector geometry defines preference distribution of firing rates: a replication study

### Models

All arm26 and point-mass effectors used to produce *Figure 5* were as described in sections 'Arm26 model' and 'Point-mass model', respectively. For all models, the policy was as described in section 'Policy network architecture', with the GRU layer containing $n = 90$ units.

### Training

All models were trained with the loss described in *Equation 1*, using a kernel regularization $\lambda = 10e^{-6}$, coefficients $\alpha = 2, \beta = 5, \gamma = 0.1, \kappa = 0.05$, and target radius $r = 0$. The models were trained on 38,400 batches of size 64, on simulations of 800 ms. The training procedure was as described in section 'General training design'.

### Testing

The testing procedure consisted of eight centre-out reaches in 800 ms simulations. Simulations started from a fixed position at a shoulder and elbow angle of 45° and 90° for the arm26 models, and at an $(x = 0, y = 0)$ cartesian position for the point-mass models. Reaches were to 24 target positions 10 cm away and distributed in increments of 15° around the starting position (*Figure 6b*). For all testing simulations, the go-cue time was fixed at 100 ms into the simulation and no catch trials were employed.

### Analysis

To obtain the PMD of each GRU, we averaged each unit's hidden activity in a 150-ms time window starting when the go-cue was input to the network (i.e., following visual feedback delay $\Delta_v$) for each reaching direction independently, and regressed that average to a diagonal design matrix encoding the reach direction. The absolute value of the resulting regression coefficients was then normalized between 0 and 1, and neurons were sorted according to these normalized coefficients to produce *Figure 6e*.

As mentioned in the results section, we trained eight networks to control an arm26 and eight networks to control a point-mass. For each network, we took the count of GRUs whose normalized regression coefficient is maximal for each target considered and averaged that count across all eight networks to produce *Figure 6c*.

## Acknowledgements

This work was supported by the Natural Science and Engineering Council of Canada (RGPIN-2018-05458 to PLG and RGPIN-2022-04421 to JAP) and the Canadian Institutes of Health Research (PJT-156241 to PLG, PJT-175010 to JAP). JAM was supported by a Banting Postdoctoral Fellowship, a BrainsCAN Postdoctoral Fellowship, and a Vector Institute Postgraduate Affiliate Program Stipend.

## Additional information

### Competing interests

J Andrew Pruszynski: Reviewing editor, *eLife*. The other authors declare that no competing interests exist.

### Funding

| Funder | Grant reference number | Author |
| --- | --- | --- |
| Natural Sciences and Engineering Research Council of Canada | RGPIN/05458-2018 | Paul L Gribble |
| Canada Research Chairs | | J Andrew Pruszynski |
| Banting Research Foundation | | Jonathan A Michaels |
| Canadian Institutes of Health Research | PJT-175010 | J Andrew Pruszynski |
| Natural Sciences and Engineering Research Council of Canada | RGPIN-2022-04421 | J Andrew Pruszynski |
| Canadian Institutes of Health Research | PJT-156241 | Paul L Gribble |

The funders had no role in study design, data collection, and interpretation, or the decision to submit the work for publication.

### Author contributions

Olivier Codol, Conceptualization, Data curation, Software, Formal analysis, Validation, Investigation, Visualization, Methodology, Writing – original draft, Project administration, Writing – review and editing; Jonathan A Michaels, Conceptualization, Data curation, Software, Formal analysis, Validation, Investigation, Visualization, Methodology, Writing – original draft, Writing – review and editing; Mehrdad Kashefi, Validation, Investigation, Visualization, Methodology, Writing – review and editing; J Andrew Pruszynski, Resources, Supervision, Validation, Writing – review and editing; Paul L Gribble, Conceptualization, Resources, Formal analysis, Supervision, Funding acquisition, Validation, Investigation, Methodology, Project administration, Writing – review and editing

### Author ORCIDs

Olivier Codol ⓘ https://orcid.org/0000-0003-0796-5457
Jonathan A Michaels ⓘ https://orcid.org/0000-0002-5179-3181
Mehrdad Kashefi ⓘ https://orcid.org/0000-0001-5981-5923
J Andrew Pruszynski ⓘ https://orcid.org/0000-0003-0786-0081
Paul L Gribble ⓘ https://orcid.org/0000-0002-1368-032X

Reviewer #1 (Public Review): https://doi.org/10.7554/eLife.88591.4.sa1
Reviewer #2 (Public Review): https://doi.org/10.7554/eLife.88591.4.sa2
Author response https://doi.org/10.7554/eLife.88591.4.sa3

## Additional files

### Supplementary files
• MDAR checklist

### Data availability

Code and sample simulations are provided in a GitHub repository here: https://github.com/OlivierCodol/MotorNet copy archived at *Codol, 2024*. Documentation for MotorNet is provided here: https://www.motornet.org.

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
