## [Editor Report · eLife assessment]

This work will be of interest to the motor control community as well as neuroAI researchers interested in how bodies constrain neural circuit function. The authors present "MotorNet", a **useful** software package to train artificial neural networks to control a biomechanical model of an effector. The manuscript provides **solid** evidence that MotorNet is easy to use and can reproduce past results in the field, both at the neural and behavioural levels. Validation is limited to planar arm-like plants or point-masses, so future work exploring three-dimensional movements and other types of plants would strengthen the impact of the tool.

---

## [Referee Report · Reviewer #1 (Public Review)]

Summary:

Codol et al. present a toolbox that allows simulating biomechanically realistic effectors and training Artificial Neural Networks (ANNs) to control them. The paper provides a detailed explanation of how the toolbox is structured and several examples demonstrating its utility.

Main comments:

(1) The paper is well-written and easy to follow. The schematics facilitate understanding of the toolbox's functionality, and the examples give insight into the potential results users can achieve.

(2) The toolbox's latest version, developed in PyTorch, is expected to offer greater benefits to the community.

(3) The new API, being compatible with Gymnasium, broadens the toolbox's application scope, enabling the use of Reinforcement Learning for training the ANNs.

Impact:

MotorNet is designed to simplify the process of simulating complex experimental setups, enabling the rapid testing of hypotheses on how the brain generates specific movements. Implemented in PyTorch and compatible with widely-used machine learning toolboxes, including Gymnasium, it offers an end-to-end pipeline for training ANNs on simulated setups. This can greatly assist experimenters in determining the focus of their subsequent efforts.

Additional context:

The main outcome of the work, a toolbox, is supplemented by a GitHub repository and a documentation webpage. Both the repository and the webpage are well-organized and user-friendly. The webpage guides users through the toolbox installation process, as well as the construction of effectors and Artificial Neural Networks (ANNs).

---

## [Referee Report · Reviewer #2 (Public Review)]

MotorNet aims to provide a unified interface where the trained RNN controller exists within the same TensorFlow environment as the end effectors being controlled. This architecture provides a much simpler interface for the researcher to develop and iterate through computational hypotheses. In addition, the authors have built a set of biomechanically realistic end effectors (e.g., a 2 joint arm model with realistic muscles) within TensorFlow that are fully differentiable.

MotorNet will prove a highly useful starting point for researchers interested in exploring the challenges of controlling movement with realistic muscle and joint dynamics. The architecture features a conveniently modular design and the inclusion of simpler arm models provides an approachable learning curve. Other state-of-the-art simulation engines offer realistic models of muscles and multi-joint arms and afford more complex object manipulation and contact dynamics than MotorNet. However, MotorNet's approach allows for direct optimization of the controller network via gradient descent rather than reinforcement learning, which is a compromise currently required when other simulation engines (as these engines' code cannot be differentiated through).

The paper has been reorganized to provide clearer signposts to guide the reader. Importantly, the software has been rewritten atop PyTorch which is increasingly popular in ML and computational neuroscience research.

---

## [Author Response]

The following is the authors’ response to the previous reviews.

**Reviewer #1 (Public Review):**
Summary:Codol et al. present a toolbox that allows simulating biomechanically realistic effectors and training Artificial Neural Networks (ANNs) to control them. The paper provides a detailed explanation of how the toolbox is structured and several examples that demonstrate its usefulness.Main comments:(1) The paper is well written and easy to follow. The schematics help in understanding how the toolbox works and the examples provide an idea of the results that the user can obtain.

We thank the reviewer for this comment.

(2) As I understand it, the main purpose of the paper should be to facilitate the usage of the toolbox. For this reason, I have missed a more explicit link to the actual code. As I see it, researchers will read this paper to figure out whether they can use MotorNet to simulate their experiments, and how they should proceed if they decide to use it. I'd say the paper provides an answer to the first question and assures that the toolbox is very easy to install and use. Maybe the authors could support this claim by adding "snippets" of code that show the key steps in building an actual example.

This is an important point, which we also considered when writing this paper. We instead decided to focus on the first approach, because it is easier to illustrate the scientific use of the toolbox using code or interactive (Jupyter) notebooks than a publication format. We find the “how to proceed” aspect of the toolbox can more easily and comprehensively be covered using online, interactive tutorials. Additionally, this allows us to update these tutorials as the toolbox evolves over different versions, while it is more difficult to update a scientific article. Consequently, we explicitly avoided code snippets on the article itself. However, we appreciate that the paper would gain in clarity if this was more explicitly stated early. We have modified the paper to include a pointer to where to find tutorials online. We added this at the last paragraph of the introduction section:

The interested reader may consult the full API documentation, including interactive tutorials on the toolbox website at https://motornet.org.

(3) The results provided in Figures 1, 4, 5 and 6 are useful, because they provide examples of the type of things one can do with the toolbox. I have a few comments that might help improving them:a. The examples in Figures 1 and 5 seem a bit redundant (same effector, similar task). Maybe the authors could show an example with a different effector or task? (see point 4).

The effectors from figures 1 and 5 are indeed very similar. However, the tasks in figure 1 and 5 present some important differences. The training procedure in figure 1 never includes any perturbations, while the one from figure 5 includes a wide range of perturbations of different magnitudes, timing and directions. The evaluation procedure of figure 1 includes center-out reaches with permanent viscous (proportional to velocity) external dynamics, while that of figure 5 are fixed, transient, square-shaped perturbation orthogonal to the reach direction. Finally, the networks in figure 1 undergo a second training procedure after evaluation while the network of figure 5 do not.

While we agree that some variation of effectors would be beneficial, we do show examples of a point-mass effector in figure 6. Overall, figure 5 shows a task that is quite different from that of figure 1 with a similar effector, while the opposite is true for figure 6. We have modified the text to clarify this for the reader, by adding the following.

End of 1st paragraph, section 2.4.

Therefore, the training protocol used for this task largely differed from section 2.1 in that the networks are exposed to a wide range of mechanical perturbations with varying characteristics.

1st paragraph of section 2.5[…] this asymmetrical representation of PMDs during reaching movements did not occur when RNNs were trained to control an effector that lacked the geometrical properties of an arm such as illustrated in Figure 4c-e and section 2.1.b. I missed a discussion on the relevance of the results shown in Figure 4. The moment arms are barely mentioned outside section 2.3. Are these results new? How can they help with motor control research?

We thank the reviewer for this comment. This relates to a point from reviewer 2 indicating that the purpose of each section was sometimes difficult to grasp as one reads. Section 2.3 explains the biomechanical properties that the toolbox implements to improve realism of the effector. They are not new results in the sense that other toolboxes implement these features (though not in differentiable formats) and these properties of biological muscles are empirically well-established. However, they are important to understand what the toolbox provides, and consequently what constraints networks must accommodate to learn efficient control policies. An example of this is the results in figure 6, where a simple effector versus a more biomechanically complex effector will yield different neural representations.

Regarding the manuscript itself, we agree that more clarity on the goal of every paragraph may improve the reader’s experience. Consequently, we ensured to specify such goals at the start of each section. Particularly, we clarify the purpose of section 2.3 by adding several sentences on this at the end of the first paragraph in that section. We also now clearly state the purpose of section 2.3 with the results of figure 6 and reference figure 4 in that section.

c. The results in Figure 6 are important, since one key asset of ANNs is that they provide access to the activity of the whole population of units that produces a given behavior. For this reason, I think it would be interesting to show the actual "empirical observations" that the results shown in Fig. 6 are replicating, hence allowing a direct comparison between the results obtained for biological and simulated neurons.

These empirical observations are available from previous electrophysiological and modelling work. Particularly, polar histograms across reaching directions like panel C are displayed in figures 2 and 3 of Scott, Gribble, Graham, Cabel (2001, Nature). Colormaps of modelled unit activity across time and reaching directions like panel F are also displayed in figure 2 of Lillicrap, Scott (2013, Neuron). Electrophysiological recordings of M1 neurons during a similar task in non-human primates can also be seen on “Preserved neural population dynamics across animals performing similar behaviour” figure 2 B (https://doi.org/10.1101/2022.09.26.509498) and “Nonlinear manifolds underlie neural population activity during behaviour” figure 2 B as well (https://doi.org/10.1101/2023.07.18.549575). Note that these two pre-prints use the same dataset.

We have added these citations to the text and made it explicit that they contain visualizations of similar modelling and empirical data for comparison:

This heterogeneous set of responses matches empirical observations in non-human primate primary motor cortex recordings (Churchland & Shenoy, 2007; Michaels et al., 2016) and replicate similar visualizations from previously published work (Fortunato et al., 2023; Lillicrap & Scott, 2013; Safaie et al., 2023).

(4) All examples in the paper use the arm26 plant as effector. Although the authors say that "users can easily declare their own custom-made effector and task objects if desired by subclassing the base Plant and Task class, respectively", this does not sound straightforward. Table 1 does not really clarify how to do it. Maybe an example that shows the actual code (see point 2) that creates a new plant (e.g. the 3-joint arm in Figure 7) would be useful.

Subclassing is a Python process more than a MotorNet process, as python is an object-oriented language. Therefore, there are many Python tutorials on subclassing in the general sense that would be beneficial for that purpose. We have amended the main text to ensure that this is clearer to the reader.

Subclassing a MotorNet object, in a more specific sense, requires overwriting some methods from the base MotorNet classes (e.g., Effector or Environment classes, which correspond to the original Plant and Task object, respectively). Since we made the decision (mentioned above) to not include code in the main text, we added tutorials to the online documentation, which include dedicated tutorials for MotorNet class subclassing. For instance, this tutorial showcases how to subclass Environment classes:

https://colab.research.google.com/github/OlivierCodol/MotorNet/blob/master/examples/3-environments.ipynb

(5) One potential limitation of the toolbox is that it is based on Tensorflow, when the field of Computational Neuroscience seems to be, or at least that's my impression, transitioning to pyTorch. How easy would it be to translate MotorNet to pyTorch? Maybe the authors could comment on this in the discussion.

We have received a significant amount of feedback asking for a PyTorch implementation of the toolbox. Consequently, we decided to enact this, and the next version of the toolbox will be exclusively in PyTorch. We will maintain the Application Programming Interface (API) and tutorial documentation for the TensorFlow version of the toolbox on the online website. However, going forward we will focus exclusively on bug-fixing and expanding from the latest version of MotorNet, which will be in PyTorch. We now believe that the greater popularity of PyTorch in the academic community makes that choice more sustainable while helping a greater proportion of research projects.

These changes led to a significant alteration of the MotorNet structure, which are reflected by changes made throughout the manuscript, notably in Figure 3 and Table 1.

(6) Supervised learning (SL) is widely used in Systems Neuroscience, especially because it is faster than reinforcement learning (RL). Thus providing the possibility of training the ANNs with SL is an important asset of the toolbox. However, SL is not always ideal, especially when the optimal strategy is not known or when there are different alternative strategies and we want to know which is the one preferred by the subject. For instance, would it be possible to implement a setup in which the ANN has to choose between 2 different paths to reach a target? (e.g. Kaufman et al. 2015 eLife). In such a scenario, RL seems to be a more natural option Would it be easy to extend MotorNet so it allows training with RL? Maybe the authors could comment on this in the discussion.

The new implementation of MotorNet that relies on PyTorch is already standardized to use an API that is compatible with Gymnasium. Gymnasium is a standard and popular interfacing toolbox used to link RL agents to environments. It is very well-documented and widely used, which will ensure that users who wish to employ RL to control MotorNet environments will be able to do so relatively effortlessly. We have added this point to accurately reflect the updated implementation, so users are aware that it is now a feature of the toolbox (new section 3.2.4.).

Impact:MotorNet aims at simplifying the process of simulating complex experimental setups to rapidly test hypotheses about how the brain produces a specific movement. By providing an end-to-end pipeline to train ANNs on the simulated setup, it can greatly help guide experimenters to decide where to focus their experimental efforts.Additional context:Being the main result a toolbox, the paper is complemented by a GitHub repository and a documentation webpage. Both the repository and the webpage are well organized and easy to navigate. The webpage walks the user through the installation of the toolbox and the building of the effectors and the ANNs.
**Reviewer #2 (Public Review):**
MotorNet aims to provide a unified interface where the trained RNN controller exists within the same TensorFlow environment as the end effectors being controlled. This architecture provides a much simpler interface for the researcher to develop and iterate through computational hypotheses. In addition, the authors have built a set of biomechanically realistic end effectors (e.g., an 2 joint arm model with realistic muscles) within TensorFlow that are fully differentiable.MotorNet will prove a highly useful starting point for researchers interested in exploring the challenges of controlling movement with realistic muscle and joint dynamics. The architecture features a conveniently modular design and the inclusion of simpler arm models provides an approachable learning curve. Other state-of-the-art simulation engines offer realistic models of muscles and multi-joint arms and afford more complex object manipulation and contact dynamics than MotorNet. However, MotorNet's approach allows for direct optimization of the controller network via gradient descent rather than reinforcement learning, which is a compromise currently required when other simulation engines (as these engines' code cannot be differentiated through).The paper could be reorganized to provide clearer signposts as to what role each section plays (e.g., that the explanation of the moment arms of different joint models serves to illustrate the complexity of realistic biomechanics, rather than a novel discovery/exposition of this manuscript). Also, if possible, it would be valuable if the authors could provide more insight into whether gradient descent finds qualitatively different solutions to RL or other non gradient-based methods. This would strengthen the argument that a fully differentiable plant is useful beyond improving training time / computational power required (although this is a sufficiently important rationale per se).

We thank the reviewer for these comments. We agree that more clarity on the section goals may improve the reader’s experience and ensured this is the case throughout the manuscript. Particularly, we added the following on the first paragraph of section 2.3, for which an explicit goal was most missing:

In this section we illustrate some of these biomechanical properties displayed by MotorNet effectors using specific examples. These properties are well-characterised in the biology and are often implemented in realistic biomechanical simulation software.

Regarding the potential difference in solutions obtained from reinforcement or supervised learning, this would represent a non-trivial amount of work to do so conclusively and so may not be within the scope of the current article. We do appreciate however that in some situations RL may be a more fitting approach to a given task design. In relation to this point we now specify in the discussion that the new API can accommodate interfacing with reinforcement learning toolboxes for those who may want to pursue this type of policy training approach when appropriate (new section 3.2.4.).

**Reviewer #3 (Public Review):**
Artificial neural networks have developed into a new research tool across various disciplines of neuroscience. However, specifically for studying neural control of movement it was extremely difficult to train those models, as they require not only simulating the neural network, but also the body parts one is interested in studying. The authors provide a solution to this problem which is built upon one of the main software packages used for deep learning (Tensorflow). This allows them to make use of state-of-the-art tools for training neural networks.They show that their toolbox is able to (re-)produce several commonly studied experiments e.g., planar reaching with and without loads. The toolbox is described in sufficient detail to get an overview of the functionality and the current state of what can be done with it. Although the authors state that only a few lines of code can reproduce such an experiment, they unfortunately don't provide any source code to reproduce their results (nor is it given in the respective repository).

The possibility of adding code snippets to the article is something we originally considered, and which aligns with comment two from reviewer one (see above). Hopefully this provides a good overview of the motivation behind our choice not to add code to the article.

The modularity of the presented toolbox makes it easy to exchange or modify single parts of an experiment e.g., the task or the neural network used as a controller. Together with the open-source nature of the toolbox, this will facilitate sharing and reproducibility across research labs.I can see how this paper can enable a whole set of new studies on neural control of movement and accelerate the turnover time for new ideas or hypotheses, as stated in the first paragraph of the Discussion section. Having such a low effort to run computational experiments will be definitely beneficial for the field of neural control of movement.

We thank the reviewer for these comments.